# Long-Term Monitoring of Extremely Low Frequency Magnetic Fields in Electric Vehicles

**DOI:** 10.3390/ijerph16193765

**Published:** 2019-10-07

**Authors:** Lei Yang, Meng Lu, Jun Lin, Congsheng Li, Chen Zhang, Zhijing Lai, Tongning Wu

**Affiliations:** China Academy of Information and Communications Technology, No.52, Huayuan bei Road, Beijing 100191, China

**Keywords:** electric vehicle, extremely low frequency, magnetic field exposure, magnetic flux density, measurement

## Abstract

Extremely low frequency (ELF) magnetic field (MF) exposure in electric vehicles (EVs) has raised public concern for human health. There have been many studies evaluating magnetic field values in these vehicles. However, there has been no report on the temporal variation of the magnetic field in the cabin. This is the first study on the long-term monitoring of actual MFs in EVs. In the study, we measured the magnetic flux density (B) in three shared vehicles over a period of two years. The measurements were performed at the front and rear seats during acceleration and constant-speed driving modes. We found that the B amplitudes and the spectral components could be modified by replacing the components and the hubs, while regular checks or maintenance did not influence the B values in the vehicle. This observation highlights the necessity of regularly monitoring ELF MF in EVs, especially after major repairs or accidents, to protect car users from potentially excessive ELF MF exposure. These results should be considered in updates of the measurement standards. The ELF MF effect should also be taken into consideration in relevant epidemiological studies.

## 1. Introduction

Electric vehicles (EVs) and hybrid electric vehicles (HEVs) have gained popularity in recent years. EVs depend on electricity, and HEVs are partially fueled by electricity. Electric motors usually react quickly, so the vehicles seem more responsive compared to those with conventional engines. At present, generation of electricity may still largely depend on the consumption of conventional energy. Nevertheless, EVs can reduce exhaust emissions in urban areas, which promotes their use in populated cities.

The health effects of extremely low frequency (ELF, frequency range from 0–100 kHz [1]) magnetic field (MF) exposure in EVs and HEVs have raised public concern. EVs consist of high-power electrical machines, inverters, high-voltage power cables and batteries. Therefore, MF exposure is inevitable in the compact metallic cabin, and the internal field distribution is very complex. ELF MF exposure, identified by International Agency for Research on Cancer (IARC) as possibly carcinogenic to humans, is closely associated with the incidence of leukemia [1]. The International Commission on Non-Ionizing Radiation Protection (ICNIRP) has proposed guidelines to establish safety limits for ELF MF exposure [2]. These guidelines use induced electric field (E-field) strength (99th percentile value, E_99_) as a basic restriction, while magnetic field strength (H) and magnetic flux density (B) are designated as reference levels (for in situ measurement) in exposure compliance assessments. The reference levels of magnetic flux density generally decrease in frequency (e.g., 0.625–0.2 mT for 8–25 Hz, 0.2 mT for 25–400 Hz, 0.2–0.0267 mT for 400–3000 Hz). These guidelines were mainly based on the short-term effects. Notably, Ahlbom et al. [3] and Greenland et al. [4] indicated that yearly exposure to 50 and 60 Hz MFs exceeding 0.3–0.4 μT may result in an increased risk of childhood leukemia, although a satisfactory causal relationship has not yet been reliably demonstrated. Hence, it is necessary to evaluate the ELF MF in commercial EVs. Researchers have conducted a number of investigations on the issue [5,6,7,8,9,10,11,12,13]. Based on the measured results, researchers have assessed human exposure to electromagnetic fields by considering the morphology and the topology of the vehicle [14].

The material used in EVs has a significant effect on the magnetic field distribution in the cabin. Some manufacturers are reported to utilize covers containing specific metal elements (e.g., beryllium copper) to reduce MF exposure in the cabin. However, regular maintenance or repairs could require mounting or dismounting the components in the cabin, which may alter the shielding and the resulting ELF MF exposure. Aging due to frequent driving could also modify the shielding (e.g., use of EVs is promoted in popular carsharing programs and many shared EVs can be driven 30,000–50,000 km per year). There is significant concern about ELF MF exposure, but the issue of MF variation during long-term usage has not yet been investigated.

In this study, we measured the ELF MF in three EVs with an interval of approximately one year. The measurements took place over two years. We recorded B values at different positions in the car, under the conditions of acceleration and driving with a constant speed of 40 km/h. The results revealed that the ELF MF values in the EVs would not change significantly due to long-term driving or regular maintenance. However, major repairs could alter both the spectrum and the amplitude of ELF MF results. The study highlighted the need to evaluate ELF MF exposure during the entire lifespan of EVs.

## 2. Materials and Methods

### 2.1. Measurement Protocols

#### 2.1.1. EV Samples

Three EVs (Table 1) were selected for MF measurement. These EVs were among the best-selling models in China according to a market survey at the time of measurement. Information about each EV is listed in Table 1.

All three EVs experienced regular maintenance every 5000–6000 km during the period of measurement. This regular maintenance included evaluations of the electrified system, the lights and the tires. The EVs were refilled with brake fluid, coolant and gear oil for the decelerator. Of note, four tires and hubs were changed on EV2 at the end of 2018. EV3 underwent a major repair immediately following a rear-end collision at the end of 2017, and its head- and rear-light assemblies were changed in 2019.

#### 2.1.2. Measurement Equipment

We measured both broadband and frequency domain results. The measurement system was composed of two ELF MF meters (SEM-600, Safetytech, Beijing, China) that were connected to two ELF MF probes (LF-01, Safetytech, Beijing, China) by optic cables. The frequency range of the probe was 1–100 kHz and its dynamic range was 0.01 nT to 10 mT. All of the instruments were within their valid calibration periods during the measurements. The measurement system fulfilled the requirements of the EN 50492-2009 [15] and ICNIRP guidelines [2,16]. During measurement, the probe was held stationary while the vehicle moved. We used an acceleration sensor (LIS3LV02DL, STMicroelectronics, Geneva, Switzerland) to monitor the acceleration rate. The ELF MF probes were fixed at the center of the front and rear passenger cushions using plastic foam. According to a previous report, B intensity increases as the distance from the EV to the ground decreases [13]. The measurement configuration is shown in Figure 1.

#### 2.1.3. Measurement Specifications

All measurements were conducted at the 2 positions (front and rear seats) shown in Figure 1a. The measurement at each position lasted for 30 s, including the two driving modes, 10 s of acceleration at a rate of 2.2 m/s^2^ (from 0–40 km/h) and 20 s of driving at a speed of 40 km/h (with a permissible variation up to ±15%). At each time point, each EV was measured for a total of ten times with broadband and frequency domain modes, respectively. During the broadband measurements, the meter took samples with three mutually orthogonal sensing coils and reported the B vector components and total B values. We configured the sampling time for the broadband meter to be 1 s. The frequency domain measurement reported the spectral components (SCs) of B components per 1 s.

Most available studies reported a spectrum of EV below 2 kHz [7,17,18]. Similar results were found in our previous measurement [13]. In the present research, we also focused on this frequency band (for both the broadband and the frequency domain measurements).

#### 2.1.4. Test Environment

The experiments were performed on Shuguang West Road in Chaoyang District, Beijing. The length of the road for the driving-test was 1.5 km (Figure 2), and the elevation varied within 5 m. The road was generally straight with relatively low automobile traffic flow (0.1–0.4 vehicles passing per second during the experiments). We conducted measurements three times on 1 August 2017, 4 August 2018 and 25 July 2019. The temperature during the measurements ranged from 32–36 °C, while humidity fluctuated from 30% to 60%. There were no high-voltage power lines along the road. The mean background field strength was 0.03 μT (broadband value: 1–2 kHz) while the peak value was less than 0.1 μT. During the experiments, municipal traffic regulations were respected.

### 2.2. Data Analysis

The purpose of the statistical study was to determine if long-term driving or repairs modify the ELF MF in EVs. For this purpose, we compared the total B strength and its SCs.

B values were measured per 1 s and averaged to obtain the results for each scenario during 10 measurement trials at three different time points. The results were used for the following statistical analysis. We performed two analyses of variance (ANOVA) to assess the difference in B values due to various factors. The first two-way repeated-measures ANOVA considered two levels of seat position (front seat and rear seat) as factors, and the measured B results (three levels: results from 2017, 2018 and 2019). The second two-way repeated-measures ANOVA considered two driving scenarios (acceleration and constant-speed driving) as factors, and the measured B results (three levels: results from 2017, 2018 and 2019). The Bonferroni correction was applied to minimize the likelihood of a type I error. Version 21.0 of the SPSS software package (IBM, Endicott, NY, USA) was used in the study. The statistical analyses were performed per vehicle.

We also analyzed the spectral components. The field meter reported spectral components per 1 s, and the first three principal SCs were selected. As a consequence, the average SCs for each measurement could be calculated for segments of 1 s.

## 3. Results

EVs used in the study were designated EV1 to EV3 to protect the commercial interest of the manufacturers.

### 3.1. Measured Broadband Results

The detailed broadband results for each measured position of three vehicles are reported in Table 2 for the three time points.

### 3.2. Statistical Analysis for the Broadband Values

No significant interaction was found between seat position and measured B results for any of the three cars (EV1: F = 0.078, *p* = 0.925; EV2: F = 0.034, *p* = 0.967; and EV3: F = 0.060, *p* = 0.942). The results for the front and the rear seat measurements revealed no difference between the surveyed vehicles (EV1: F = 0.235, *p* = 0.629; EV2: F = 0.005, *p* = 0.944; and EV3: F = 0.014, *p* = 0.907). In contrast, we observed significant differences between the measured B results for EV2 (F = 0.129, *p* = 0.006) and EV3 (F = 17.76, *p* < 0.001), but not EV1 (F = 0.129, *p* = 0.879). Upon detecting this difference, we conducted multiple comparisons. The measured B values for EV2 from 2017 were significantly different from the results from 2019 (*p* = 0.009 by Bonferroni correction) and 2018 (*p* = 0.03 from Bonferroni correction). EV3 also showed a substantial difference between the B results from different measurement time points (*p* < 0.016 from Bonferroni correction). These results indicate that the measured B strength changed significantly each year.

The other two-way repeated-measures ANOVA (driving scenario × B value) revealed significant interactional differences between the three vehicles (EV1: *p* = 0.003; EV2: *p* = 0.001 and EV3: *p* = 0.001). Among the two factors, acceleration differed significantly from the constant-speed driving (EV1: F = 4643, *p* < 0.001; EV2: F = 3200, *p* < 0.001 and EV3: F = 1331, *p* < 0.001), which was confirmed by previous reports, because acceleration needs more traction power and high current, which is associated with greater B strength [13]. The same effect has been detected regarding measured B strength. EV1 had no significant difference in year-to-year results. For EV2, B value results from 2019 were obviously different from the results from 2018 (*p* = 0.001 from Bonferroni correction) and 2017 (*p* = 0.001 from Bonferroni correction), while the results from those two years were similar (*p* = 0.06 from Bonferroni correction). For EV3, significant differences were found between the results from any two years (*p* < 0.001 from Bonferroni correction).

### 3.3. Frequency Domain Results

The spectral results were analyzed. The first three major spectral components (SCs, the three frequency components which have the highest amplitude were sorted by amplitude) for the measured results are plotted in Figure 3 (acceleration) and Figure 4 (constant-speed driving). The SC frequency volatility apparent in the figures may be due to measurement uncertainty and substantial modification of the spectrum during operation of the electrified systems. For EV3, the variation between the major SCs was larger than the range for EV1. There was a similar trend between the last measurement of EV2 and its previous results. For EV1, the variation between SCs during the two years was rather small. The maximum variation of B for EV3, which underwent major repairs and component replacement, was up to 0.54 μT in the acceleration mode (difference between 2019SC1 and 2017SC1 for front seat position), whereas the maximum variation for EV1, which only underwent regular maintenance, was approximately 0.02 μT (difference between 2019SC1 and 2017SC1 for front seat position). EV2, which had its hubs changed, had a variation between those of EV1 and EV3. The results for the constant-driving mode revealed the same effect. In general, the frequency domain results were consistent.

## 4. Discussion

The ELF MF was produced through the operation of the EV powertrain, which correlated with the actual output power. Hence power, weight, acceleration and velocity may influence the B strength in the cabin. The measured results generally fell in the range of about several tenths of μT, which is consistent with previous studies [13] measured under similar conditions. The measured results were validated. Previous studies focused on evaluating the magnetic field distribution in the cabin of EVs and the spectral analysis [7,11]. The acceleration/deceleration session usually involves higher ELF MF values. Our measurements were conducted with a moderate acceleration rate (2.2 m/s^2^) and constant-speed driving. The analysis revealed significantly higher B values for the acceleration scenario that correlated with previous reports [6,7,13]—to achieve elevated acceleration, EVs need output higher power, and a larger current introduces higher ELF MF values. Repeated measurements over 2 years confirmed the findings even with EVs different from the previous reports. A new finding can be derived from EV1, where the MF does not change even after multiple regular maintenance services.

The statistical results revealed a difference between the broadband B values measured for EV3 at the three time points, and the last time measurement for EV2, while no difference was found for EV1. This change was also observed in the frequency domain. The change in B strength corresponded with a major repair and replacement of the lights and tires. As expected, the usage of spare components, or inappropriately mounting the components during repair, could modify the performance of ELF MF shielding, resulting in a perturbed MF in the cabin. This phenomenon was the case for EV3.

Replacing tires could change the MF in the cabin. The mechanism used was the magnetization of the wire when it spun in a terrestrial MF [6]. Although the hubs of the tested vehicle were of a material with low magnetic permeability (aluminum alloy), the steel wire in the reinforcing belts of tires picked up MF from the terrestrial MF as the tires rotated. The generated ELF MF was usually below 20 Hz, but possibly exceeded 2.0 µT at the seat level in the passenger compartment [6]. We did not intentionally degauss the magnetized tires before driving to maintain a realistic exposure scenario for the occupants. This can explain the EV2 results.

A significant interaction was reported between driving scenario and the measured B. This effect may indicate that the operation of the electrified systems by acceleration produced perturbed magnetic field distributions, which enhanced the difference in B due to the replacement of the components. In the experiment, we did not evaluate the case of rapid acceleration, since the speed limit of the route was 80 km/h, and an acceleration of 2.2 m/s^2^ in 10 s reaches the limit. Note that the purpose of the study was to investigate the factors influencing the MF during long-term usage. The effect we found would not change with higher acceleration.

In the experiments, we evaluated data from three vehicles. It is difficult to track more vehicles due to the high mobility of shared cars. The cars were rented every day, and many of them were permanently moved to other cities. Nevertheless, the results were consistent, and clearly revealed that major repairs can modify the MF in the cabin, which was sufficient for this study. In future, we need to perform detailed studies on the relationship between the B difference and specific repairs/component replacement. In our case, B was enhanced, but this may be due to the specific material of the replaced component or the assembly technique (loose assembly). Further investigation should be conducted on this topic.

The measurements were not performed on the left-side seats because the left-front seat was occupied by the driver, and the left-rear sear was occupied by the field meter and acceleration sensor. The field strength measured at the left positions may vary, but the purpose of the study was to assess the trend of MF variation over long-term usage. Our finding was that major repairs and the replacement of components can modify the field in the cabin. The MF was modified in the entire cabin, and the trend would not vary with changes in position. The measurements were conducted on a city road. The environmental MF strength was monitored. In addition, the road was rather straight with little traffic. The results were minimally influenced by environmental factors.

Standards exist for evaluating the MF in a vehicle. These standards focus on instrumentation, measurement points, measurement time, postprocessing for the field values, etc. For this study, it was essential to regularly measure the MF in EVs for the purpose of epidemiological studies, because the MF distribution could vary significantly with long-term usage. A long-term survey should be conducted to confirm the findings.

## 5. Conclusions

In this study, we monitored ELF MF in three shared vehicles over two years. The measurements were performed at the front and the rear seats under acceleration and constant-driving modes. We found that the broadband B value was significantly changed with replacement of the components and the tires while regular checks or maintenance did not influence the measured B values in the vehicle. The variation of the major spectral components of B was larger for the repaired cars, compared to the results from the cars with regular maintenance. These results highlight the necessity of regularly monitoring the ELF MF in EVs, especially after major repairs or accidents, to protect car users from MF exposure.

## Figures and Tables

**Figure 1 ijerph-16-03765-f001:**
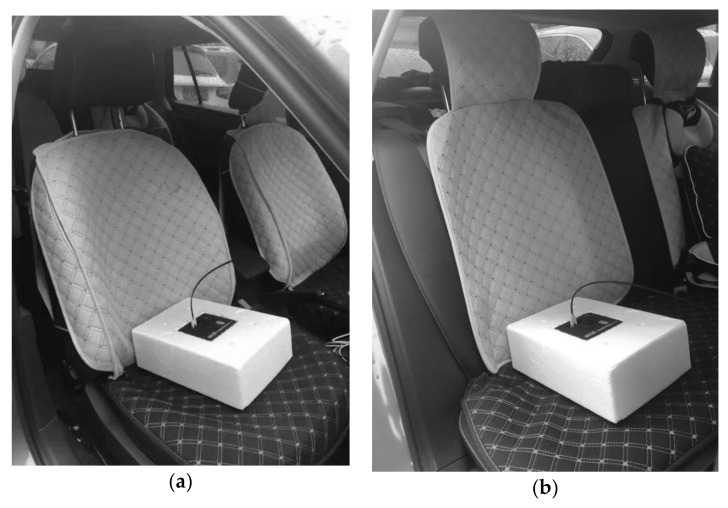
Positioning of the extremely low frequency (ELF) magnetic field (MF) probes at the right front (**a**) and rear (**b**) seats.

**Figure 2 ijerph-16-03765-f002:**
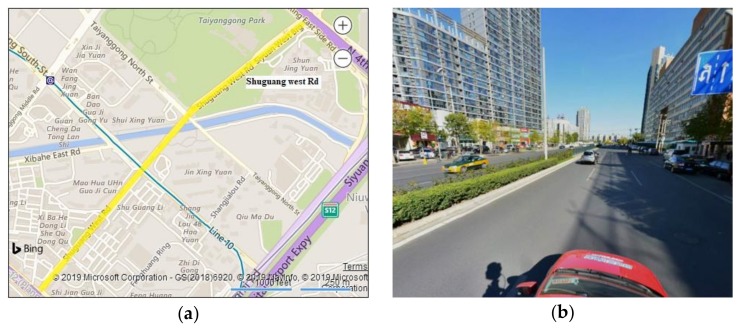
Driving test environment. (**a**) is a map of its location in Beijing and (**b**) is a picture of the road.

**Figure 3 ijerph-16-03765-f003:**
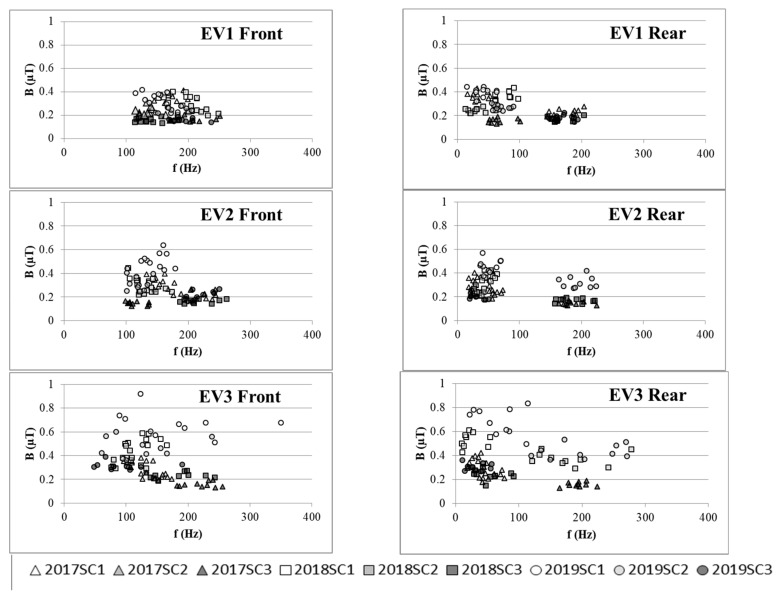
Measured spectral components (SCs) from the acceleration sessions at rear and front seats. The signs in the caption are formed of four digits representing the measurement year and three digits representing the specific SC, e.g., 2019SC3 indicating the third highest spectral component from the measurement in 2019.

**Figure 4 ijerph-16-03765-f004:**
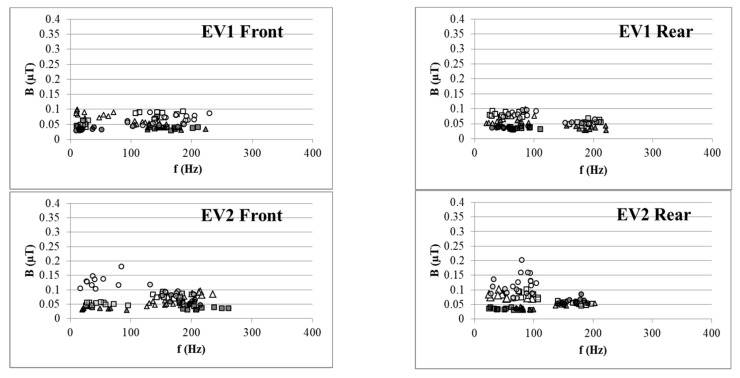
Measured SCs from the constant-driving modes at rear and front sets. The signs in the caption are formed of four digits representing the measurement year and three digits representing the specific SC, e.g., 2019SC3 indicating the third highest spectral component from the measurement in 2019.

**Table 1 ijerph-16-03765-t001:** Electric vehicles (EVs) tested. Total weight of EV includes the weight of one driver and the measurement equipment.

Manufacturer	Type	Maximal Power (kW)	Net Weight of EV (kg)	Total Weight of EV (kg)	Mileage of the 1st Test	Mileage of the 2nd Test	Mileage of the 3rd Test
BYD	QIN	160	1750	1818.2	5543 km	34,674 km	79,865 km
BYD	YUAN	70	1330	1398.2	4987 km	41,852 km	81,262 km
Beijing Automotive	EX360	80	1480	1548.2	6623 km	39,876 km	88,754 km

BYD—Name of Manufactuerer; QIN—Type of vehicle; YUAN—Type of vehicle; EX360—Type of vehicle; EV—Electric vehicle.

**Table 2 ijerph-16-03765-t002:** Measured broadband B strength. A: acceleration; C: constant-speed driving; F: front seat; R: rear seat. The results are expressed in the form of mean ± standard deviation. Units in μT.

EV 1				
		04/08/2017	03/08/2018	25/07/2019
A	F	0.79 ± 0.06	0.78 ± 0.07	0.81 ± 0.07
A	R	0.81 ± 0.07	0.82 ± 0.05	0.92 ± 0.07
C	F	0.18 ± 0.02	0.18 ± 0.01	0.17 ± 0.01
C	R	0.18 ± 0.02	0.18 ± 0.01	0.19 ± 0.01
EV 2				
		04/08/2017	03/08/2018	25/07/2019
A	F	0.74 ± 0.05	0.80 ± 0.06	1.13 ± 0.13
A	R	0.75 ± 0.05	0.84 ± 0.06	1.07 ± 0.12
C	F	0.18 ± 0.02	0.18 ± 0.02	0.30 ± 0.04
C	R	0.18 ± 0.01	0.18 ± 0.02	0.32 ± 0.06
EV 3				
		04/08/2017	03/08/2018	25/07/2019
A	F	0.79 ± 0.06	1.14 ± 0.13	1.57 ± 0.21
A	R	0.76 ± 0.07	1.18 ± 0.16	1.52 ± 0.15
C	F	0.18 ± 0.01	0.34 ± 0.09	0.58 ± 0.11
C	R	0.18 ± 0.02	0.35 ± 0.09	0.55 ± 0.19

A—acceleration; C—constant-speed driving; F—front seat; R—rear seat.

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
