# Peer review of "Long-Term Monitoring of Extremely Low Frequency Magnetic Fields in Electric Vehicles"

_ijerph, 2019, doi:10.3390/ijerph16193765_

Round 1

Reviewer 1 Report

This work evaluates the magnetic field exposure in three EVs with an interval of approximate one year..

The work is correct but you should take into account the following recommnedations:

1.- You should resume the different sections fo the paper at the end of the Introduction section

2.- The interest of the topic of work should be highlighted in the Introduction. You should remark the relevance of the study of human exposure in vehicles [*]:

[*] Celaya-Echarri, M; Azpilicueta, L; Lopez-Iturri, P; Aguirre, E; De Miguel-Bilbao, S; Ramos, V; Falcone, F. "Spatial Characterization of Personal RF-EMF Exposure in Public Transportation Buses," in IEEE Access, vol. 7, pp. 33038-33054, 2019.

3.- You should remark the factors that affect the magnetic field exposure in de Discussion section. You should contrast your conclusions with previous bibliography.

On the other hand, as for the mode of writing the work is correct, and it is well understood

Author Response

This work evaluates the magnetic field exposure in three EVs with an interval of approximate one year.

The work is correct but you should take into account the following recommnedations:

1.- You should resume the different sections fo the paper at the end of the Introduction section

-- We resumed the different sections at the end of the introduction “This paper was organized as follows: in Section 2, the EVs used for measurement, measurement equipment, measurement protocol, test environment, and data analysis methods were presented. In Section 3, both the broadband and the frequency domain results were presented and analyzed. Section 4 focused on a discussion of the results in terms of the representation and validity of the results and the relation of statistical B variation to maintenance and repair. Conclusions were presented in Section 5. ”

2.- The interest of the topic of work should be highlighted in the Introduction. You should remark the relevance of the study of human exposure in vehicles [*]:

[*] Celaya-Echarri, M; Azpilicueta, L; Lopez-Iturri, P; Aguirre, E; De Miguel-Bilbao, S; Ramos, V; Falcone, F. "Spatial Characterization of Personal RF-EMF Exposure in Public Transportation Buses," in IEEE Access, vol. 7, pp. 33038-33054, 2019.

-- Thanks for the advice. This reference is new and important. We added in the manuscript.

3.- You should remark the factors that affect the magnetic field exposure in de Discussion section. You should contrast your conclusions with previous bibliography. On the other hand, as for the mode of writing the work is correct, and it is well understood

 -- Thanks for the suggestion. We added the information at the beginning of the discussion part “The MF was produced by the operation of the EV powertrain, which was correlated with the actual output power. Hence, the power, weight, acceleration and velocity may influence the B strength in the cabin. The measured results generally fell in the range of about several tenths of μT, which is consistent with previous studies [13] measured at the similar conditions. The measured results were validated. Previous studies focused on evaluating the magnetic field distribution in the cabin of EVs and the spectral analysis [19,20]. The acceleration/deceleration session usually involves higher MF values. Our measurements were conducted with a moderate acceleration rate (2.2 m/s2) and constant-speed driving. The analysis revealed a significantly higher B for the acceleration scenario that correlated with previous reports [6,7,13], and the explanation is reasonable; to achieve elevated acceleration, EVs need output higher power, and larger current introduces higher MF. The repeated measurements over 2 years confirmed the finding even with EVs different from the previous reports. A new finding can be derived from EV1 that the MF does not change even after multiple regular maintenance services.” 

Reviewer 2 Report

The Article "Chronic Monitoring of Extremely low Frequency Magnetic Fields in Electric Vehicles" describes repeated measurements of magnetic fields in electric vehicles, and the variation of these measures over time, when the vehicle is routinely maintained  or subject to some repair. 

The measurement results  corroborate the suggestion to regularly monitor magnetic fields in electric vehicles, because the shielding and the MF-exposure for humans on the seats may change during the lifetime of the vehicle in particular after repair work.  

I consider it a pitty that only three vehicles were involved, so it describes a principle and not scientific evaluation, that could be representative.  

In the following I want to provide some reflections to - in my opinion - make the MS stronger: 

line 13: "... during 3 years ... " the 3 measurements were during 2 years only, the 3 measurements were one year apart from each other. Please sort this out. 

line 26: "They  are ... partially fuelled with electricity. Who is they? EVs or HEVs,

then: is fueled the proper term to say they are operated with electric energy? 

line 31: "... the health effect ... range from ... 0 to 100 kHz ... "   Please sort this out,  there is no kHz unit for health effect ranges.   

line 35: " ... complicated ...?" do you mean "complex" ? 

line 54-55: "There is an important concern ... not yet investigated"  I do not understand this sentences here. Was it the concern which is not investigated, or was something of concern not investigated? please sort this out. 

line 57: "The measurement lastet for two years .... ", Please sort out the contradiction in line 13. (two years / three years)

line 92 - 94: are the "two driving sessions" two driving modes (acceleration and  steady speed)?  Does "measurement of each point" refer to each of the three time points over the 2 years? And did each of the 10 measurements  at one point, consist of 1.) 10 sec acceleration followed by 2.) 20 constant speed driving at 40/km/h?. Then  at each point the vehicle was accellerated 10 times to record and collect all data, correct? Can you make this more comprehensible please? 

line 117: The 1-s measured values were averaged .... . 2 Issues here: 1.) What is "1-s" or does it refer to the sampling time "1 s" ? 2.) presumable the averaged 1s sampling values are the values in table 1, if so - please confirm. 

line 118: consequent ? - Consecutive ? 

line 132 - 133, this information should go to Table1, e.g.  line 70 before the terms  EV1, EV2, EV3  are used.  

line 137, Table 2: What is the point of showing the raw data, instead of the mean ± STD, N=10.    the comparison between the vehicles would gain a lot and become much more instructive if the average of the 10 samples would be included in an additional  column? or the raw data can be omitted. 

line 172: concentrated ? - small. 

173 Please change the figure headings "Vehicle A" should be vehicle 1, to be consistent with rest of the paper and the legend. 

line 184: .... accelleration session (mode) was an approbriate ... worst case measurement. 
This  exposure is presumably only a small part of the overall exposure. From a Public Health perspective it would be interested to discuss the relevance of the acceleration mode MF exposure for the total exposures while using the EV.  

line 217: "... during three years ...", or was it two years and 3 time points? 

end of Comments -------------------------------------------------------

Author Response

The Article "Chronic Monitoring of Extremely low Frequency Magnetic Fields in Electric Vehicles" describes repeated measurements of magnetic fields in electric vehicles, and the variation of these measures over time, when the vehicle is routinely maintained  or subject to some repair. 

The measurement results corroborate the suggestion to regularly monitor magnetic fields in electric vehicles, because the shielding and the MF-exposure for humans on the seats may change during the lifetime of the vehicle in particular after repair work.  

I consider it a pitty that only three vehicles were involved, so it describes a principle and not scientific evaluation, that could be representative.  

In the following I want to provide some reflections to - in my opinion - make the MS stronger: 

line 13: "... during 3 years ... " the 3 measurements were during 2 years only, the 3 measurements were one year apart from each other. Please sort this out. 

--Thanks for the advice. We made corrections throughout the manuscript.

line 26: "They  are ... partially fuelled with electricity. Who is they? EVs or HEVs, then: is fueled the proper term to say they are operated with electric energy? 

--Thanks for the advice. We clarified the paragraph by “Electric vehicles (EVs) and hybrid electric vehicles (HEVs) have gained popularity in recent years. EVs depend on electricity, and HEVs are partially fueled by electricity”.We have the entire manuscript edited by Wiley editing service (Order ID: TVZWKSXW) and the expression has been completed corrected by native speakers.

line 31: "... the health effect ... range from ... 0 to 100 kHz ... "   Please sort this out,  there is no kHz unit for health effect ranges. 

--Thanks for the advice we changed it to “The health effect of extremely low frequency (ELF, frequency range from 0 to 100 kHz [1]) magnetic field (MF) exposure in EVs and HEVs has raised public concern.”  

line 35: " ... complicated ...?" do you mean "complex" ? 

--Thanks for the advice. We corrected it.

line 54-55: "There is an important concern ... not yet investigated"  I do not understand this sentences here. Was it the concern which is not investigated, or was something of concern not investigated? please sort this out. 

--Yes we need to clarify it and we changed it to “There is significant concern about ELF MF exposure, but the issue of MF variation during long-term usage has not yet been investigated.”

line 57: "The measurement lastet for two years .... ", Please sort out the contradiction in line 13. (two years / three years)

--We made corrections throughout the manuscript.it was two years.

line 92 - 94: are the "two driving sessions" two driving modes (acceleration and  steady speed)?  Does "measurement of each point" refer to each of the three time points over the 2 years? And did each of the 10 measurements  at one point, consist of 1.) 10 sec acceleration followed by 2.) 20 constant speed driving at 40/km/h?. Then  at each point the vehicle was accellerated 10 times to record and collect all data, correct? Can you make this more comprehensible please? 

--Yes. you understand well for the measurement configuration. We improved the sentences “All measurements were conducted at the 2 positions (front and rear seats) shown in Figure 1a. The measurement at each position lasted for 30 s, including the two driving modes, 10 s of acceleration at a rate of 2.2 m/s2 (from 0 to 40 km/h) and 20 s of driving at a speed of 40 km/h (with a permissible variation up to ±15%). At each time point, each EV was measured for a total of ten times with broadband and frequency domain modes, respectively.”

line 117: The 1-s measured values were averaged .... . 2 Issues here: 1.) What is "1-s" or does it refer to the sampling time "1 s" ? 2.) presumable the averaged 1s sampling values are the values in table 1, if so - please confirm. 

--Yes. We changed 1-s to 1s.

line 118: consequent ? - Consecutive ? 

--We changed the expression to “B values were measured per 1 s and averaged to obtain the results for each scenario during 10 measurement trials at three different time points. The results were used for the following statistical analysis. We performed two analyses of variance (ANOVA) to assess the difference in B strength due to various factors. The first two-way repeated-measures ANOVA considered two levels of seat position (front seat and rear seat) as factors and the measured B results (three levels: results from 2017, 2018 and 2019). The second two-way repeated-measures ANOVA considered two driving scenarios (acceleration and constant-speed driving) as factors and the measured B results (three levels: results from 2017, 2018 and 2019). The Bonferroni correction was applied to minimize the likelihood of a type I error. The two kinds of corrections were applied to show the significant values with different criteria. Version 21.0 of the SPSS software package (IBM, Endicott, NY, USA) was used in the study. The statistical analyses were performed per vehicle.”

line 132 - 133, this information should go to Table1, e.g.  line 70 before the terms  EV1, EV2, EV3  are used.  

--Yes. We made the corrections accordingly.

line 137, Table 2: What is the point of showing the raw data, instead of the mean ± STD, N=10.    the comparison between the vehicles would gain a lot and become much more instructive if the average of the 10 samples would be included in an additional  column? or the raw data can be omitted. 

--We believe that your suggestion is important. We provide the mean±std along with the raw data. We hope that the readers can use the raw data for further analysis or to repeat the results.

line 172: concentrated ? - small. 

--We changed it.

173 Please change the figure headings "Vehicle A" should be vehicle 1, to be consistent with rest of the paper and the legend. 

--We changed it.

line 184: .... accelleration session (mode) was an approbriate ... worst case measurement. 
This  exposure is presumably only a small part of the overall exposure. From a Public Health perspective it would be interested to discuss the relevance of the acceleration mode MF exposure for the total exposures while using the EV.  

--Thanks for your suggestion. We believe that the expression was not appropriate. We deleted the sentence and re-write the paragraph.

line 217: "... during three years ...", or was it two years and 3 time points? 

--It is changed to two years.

Reviewer 3 Report

This paper describes the variation of B field over two years in time and frequency domain in two seating positions in electric vehicles. The paper puts some insight in variation B field variation over time were factors such as repair could are suggested as reason to this variation. The paper, however, needs some improvements before it is ready for publication.

Overall comments:

The authors need to revisit the aim of the paper and to make sure that the presented results and conclusions are in line with this.

INTRODUCTION:

page 1 line 44: 0.3-0.4µT is originally stated as a mean value over one year, please clarify this.

page 2 line 58: the statement that the B-field is not changed significantly during regular maintenance, even though true, is not stated in the discussion or conclusion, please clarify.

MATERIAL AND METHODS:

Why did you chose the right rear/front seats, were there any B-field variations on right/left hand side?

Why did you chose to perform measurements during normal driving in a city? To find difference due to reparation it might have been easier to draw conclusions from measurements in a test environment.

The statistical method is unclear, the first repeated measure ANOVA included position as one factor and the other type of speed. Did you consider to combine these two analyses in line with the hypothesis. Where the mean value of the measured B field used in the analyses?

Why were both Bonneferoni and LSD correction used?

page 4 line 129: Define SC

RESULTS:

Table 2 could be condensed by expressing the mean value and standard deviation for each measurement instead of 10 repeated measurements. Please change the "total B-field " to broadband B-field.

page 6 line 151 and 160: just a clarification, measurement year is included as a within subject factor in the ANOVA I suppose, the wording "between year" might then be a bit confusing.

The frequency domain variation is poorly visualised. Please consider to present this differently. Once again, what is SC?

The visual difference in frequency content between rear and front in Figure 3 indicated that there are different sources to the measured B-field, is the variation over time equal between front and rear?

DISCUSSION

The discussion about how repair might affect the B-field values is vague and are based on assumptions since no clear follow up before/after repair was done. If no further information on what has been replaced on EV2-3 and not on EV1 is available I would recommend the authors to rather state what is needed to be done in further studies to explain the difference. 

The authors claims in the introduction that theses results are of importance for measurement standards and for epidemiological studies, but this is barely mentioned in the discussion.

CONCLUSION:

page 8 line 219: there are no significantly differences in frequency domain shown in the results, no statistical test were used.

line 222, It would be better to only draw conclusion based on the stated aim and results, "unnecessary exposure " is in my opinion, not relevant in the conclusion.

Author Response

This paper describes the variation of B field over two years in time and frequency domain in two seating positions in electric vehicles. The paper puts some insight in variation B field variation over time were factors such as repair could are suggested as reason to this variation. The paper, however, needs some improvements before it is ready for publication.

-- Thanks for the comments. We made corrections accordingly. 

Overall comments:

The authors need to revisit the aim of the paper and to make sure that the presented results and conclusions are in line with this.

--Thanks for the advice. We made thorough correction for the discussion and the conclusion parts to emphasize the focus of the manuscript. A summary for each section was also added at the end of introduction.

INTRODUCTION:

page 1 line 44: 0.3-0.4µT is originally stated as a mean value over one year, please clarify this.

--The expression has been changed to “Notably, Ahlbom et al. [3] and Greenland et al. [4] indicated that yearly exposure to 50 and 60 Hz MFs exceeding 0.3 - 0.4 μT may result in an increased risk for childhood leukemia, although a satisfactory causal relationship has not yet been reliably demonstrated.”

page 2 line 58: the statement that the B-field is not changed significantly during regular maintenance, even though true, is not stated in the discussion or conclusion, please clarify.

--We have added the finding in discussion “A new finding can be derived from EV1 that the MF does not change even after multiple regular maintenance services.”

MATERIAL AND METHODS:

Why did you chose the right rear/front seats, were there any B-field variations on right/left hand side?

-- The left side of the front row is driving position, so it cannot be measured in driving. There might be some difference of B due to the difference distance to the internal electrified system for various seats. In the present study, we focused our work on comparing the long-term variation of B in the vehicle especially after repair or replacing the components. We believed that our finding would not change for different seats. 

Why did you chose to perform measurements during normal driving in a city? To find difference due to reparation it might have been easier to draw conclusions from measurements in a test environment.

--We do not have a test environment dedicating to driving. The selected road was rather straight, flat and with low environmental MF. We believe that the road is close to the test road. In the discussion, we added “The measurements were not performed on the left-side seats because the left-front seat was occupied by the driver, and the left-rear sear was occupied by the field meter and acceleration sensor. The field strength measured at the left positions may vary, but the purpose of the study was to assess the trend of MF variation during long-term usage. Our finding was that major repairs and the replacement of components can modify the field in the cabin. The MF was modified in the entire cabin, and the trend would not vary with position. The measurements were conducted on a city road The environmental MF strength was monitored. In addition, the road was rather straight with little traffic. The results were minimally influenced by environmental factors. ”

The statistical method is unclear, the first repeated measure ANOVA included position as one factor and the other type of speed. Did you consider to combine these two analyses in line with the hypothesis. Where the mean value of the measured B field used in the analyses?

Why were both Bonneferoni and LSD correction used?

--The number of samples was limited and it was not sufficient for three way ANOVA. As proposed by the second reviewer, mean B values should be presented and we listed them in Table 2. The samples to compare were the mean B. Yes. It is unnecessary to have both Bonneferoni and LSD correction. We keep Bonferroni correction results.

page 4 line 129: Define SC

--It is defined in line 119. It is spectral component

RESULTS:

Table 2 could be condensed by expressing the mean value and standard deviation for each measurement instead of 10 repeated measurements. Please change the "total B-field " to broadband B-field.

--Thanks for the suggestion. It has been changed throughout the manuscript.

page 6 line 151 and 160: just a clarification, measurement year is included as a within subject factor in the ANOVA I suppose, the wording "between year" might then be a bit confusing.

--Yes. We clarified the expression. We added the year in the paragraph and significantly update the section.

The frequency domain variation is poorly visualised. Please consider to present this differently. Once again, what is SC?

--We improved the figure by its resolution and the legends. The visualization style of the figures was also changed. Anyway, since the variation of the measurement points was not obvious for all the situations, we tried to make it clearer. SC is mentioned in line 119.

The visual difference in frequency content between rear and front in Figure 3 indicated that there are different sources to the measured B-field, is the variation over time equal between front and rear?

--The front and rear seats may have different distance to the internal electrified system which induced ELF MF exposure. So, if they have difference, it depends on the specific vehicle. The purpose of the manuscript was to compare the difference for long-time usage., especially after repair or replacing the components. Hence, we did not compare the field for front and rear seats.  

DISCUSSION

The discussion about how repair might affect the B-field values is vague and are based on assumptions since no clear follow up before/after repair was done. If no further information on what has been replaced on EV2-3 and not on EV1 is available I would recommend the authors to rather state what is needed to be done in further studies to explain the difference. 

--We added it in discussion and conclusion “Further investigation should be conducted on this topic.”

The authors claims in the introduction that theses results are of importance for measurement standards and for epidemiological studies, but this is barely mentioned in the discussion.

--We added at the last part of discussion “Standards exist for evaluating the MF in a vehicle. These standards focus on instrumentation, measurement points, measurement time, postprocessing for the field values, etc. For this study, it was essential to regularly measure the MF in EVs for the purpose of epidemiological studies because the MF distribution could vary significantly with long-term usage. A long-term survey should be conducted to confirm the findings.”

CONCLUSION:

page 8 line 219: there are no significantly differences in frequency domain shown in the results, no statistical test were used.

line 222, It would be better to only draw conclusion based on the stated aim and results, "unnecessary exposure " is in my opinion, not relevant in the conclusion.

--We changed the expression

Round 2

Reviewer 3 Report

The authors have made some improvements in line with both reviewers comments, but I still think more work on the manuscript is needed before it could be accepted for publication.

It seems like the manuscript has been spelling checked, but there are still some spelling and grammar issues that needs to be taken care of. For instance the authors write "B results" when I assume they mean B-field. Statistical study should be statistical analyses (line 124). "We measured both broad band and frequency domain results" (line 85) could be replaced by "We measured the B-field in both time and frequency domain." Similar problem in the abstract (line 15).

Spectral results could be replaced by spectral component.

ABSTRACT

The spectral component of the B-field is not "significantly" modifiedsince no statistical analyses on the spectral component has been applied in the results.

INTRODUCTION

There is a mixture of long term effect and regulations based on short term effects (line 33-44), please sort it out.

The added sentence (line 62-68) are in my opinion not necessary since this is common practice in scientific articles in general.

MATERIALS AND METHODS

The authors states that the measurement position is chosen to correspond to the reproductive system, but any discussion on effects on the reproductive system is not included. I suggest to remove this sentence.

Which axes (x,y,z) is the spectral component  reported for. Also show this axes in Figure 1.

I think it was a good idea to omit the LSD correction, please revisit line 134 and 171, it says two corrections still

Good that you have included the mean and SD in tale 2, but I strongly recommend that the raw data is omitted. It is space consuming and very uncommon to include this.

The analyses of the frequency components are still vague, what does the "first three major SC" mean and are for instance 2017SC1, 2017SC2 and 2017SC3 three SC majors?

"up to 0.9 µT" (line 179) is hard to understand in which frequency range? what dots should the reader compare in the figures.

CONCLUSION

To be able to support the thesis that the changes in B-field after repair and change of tires it would be good to support this with info on repair for each EV, for instance in Table 1.

The spectral component is not significantly changed since no statistical analyses have been applied on this comparison.

Author Response

The authors have made some improvements in line with both reviewers comments, but I still think more work on the manuscript is needed before it could be accepted for publication.

--Thanks for your advice. We will continue to improve the quality of the manuscript. We tracked the updates in the manuscript.

It seems like the manuscript has been spelling checked, but there are still some spelling and grammar issues that needs to be taken care of. For instance the authors write "B results" when I assume they mean B-field. Statistical study should be statistical analyses (line 124). "We measured both broad band and frequency domain results" (line 85) could be replaced by "We measured the B-field in both time and frequency domain." Similar problem in the abstract (line 15).

--Thanks for the advice. In the study, B indicates magnetic flux density as we defined in Introduction. To clarify, we changed B results to ELF MF results.

Spectral results could be replaced by spectral component.

--We made the corrections accordingly. It has been changed to spectral components.

ABSTRACT

The spectral component of the B-field is not "significantly" modified since no statistical analyses on the spectral component has been applied in the results.

--We deleted it.

INTRODUCTION

There is a mixture of long term effect and regulations based on short term effects (line 33-44), please sort it out.

--Thanks for the advice. We notice the authors that the guidelines were mainly based on short-term effects. The studies by Ahlbom et al and Greenland et al were from long-term surveys. We clarified it by adding “The guidelines were mainly based on the short-term effects.” before the two publications.

The added sentence (line 62-68) are in my opinion not necessary since this is common practice in scientific articles in general.

--We deleted it.

MATERIALS AND METHODS

The authors states that the measurement position is chosen to correspond to the reproductive system, but any discussion on effects on the reproductive system is not included. I suggest to remove this sentence.

--We deleted it.

Which axes (x,y,z) is the spectral component  reported for. Also show this axes in Figure 1.

--Actually, we did not fix the placement of the probe in the vehicle. Because the field in the EV was perturbed by the metallic cabin, it is unnecessary to record the field strength on specific axis.

I think it was a good idea to omit the LSD correction, please revisit line 134 and 171, it says two corrections still

--Thanks for the advice. We deleted them.

Good that you have included the mean and SD in tale 2, but I strongly recommend that the raw data is omitted. It is space consuming and very uncommon to include this.

--Thanks for your advice. We have carefully considered it. The authors have discussed on your suggestion. In view of the integrity of the study, we hope to keep the raw data. We understand your concern, so if the journal editor asks us to remove them due to the limited space of the publication, we are also willing to following the instruction.

The analyses of the frequency components are still vague, what does the "first three major SC" mean and are for instance 2017SC1, 2017SC2 and 2017SC3 three SC majors?

--We made clarification by adding “The spectral results were analyzed. The first three major SCs (the three frequency components which have the highest amplitude, were sorted by amplitude) for the measured results are plotted in Figure 3 (acceleration) and Figure 4 (constant-speed driving)..” At the beginning of section 3.3. We also added the explanation in the caption of the figure “The signs in the caption are formed by first four digits representing the measurement year and three digits representing the specific SC, e.g., 2019SC3 indicating the third highest spectral component from the measurement in 2019.” We hope that will make the manuscript much clearer.

"up to 0.9 µT" (line 179) is hard to understand in which frequency range? what dots should the reader compare in the figures.

--Thanks for the suggestion. We made a mistake here. We corrected and clarified it by “The spectral results were analyzed. The first three major SCs (the three frequency components which have the highest amplitude, were sorted by amplitude) for the measured results are plotted in Figure 3 (acceleration) and Figure 4 (constant-speed driving). We summarized the results in Table 3. The SC frequency volatility apparent in the figures may be due to measurement uncertainty and substantial modification of the spectrum during operation of the electrified systems. For EV3, the range of variation of the major SCs was larger than the range for EV1. There was a similar trend between the last measurement of EV2 and its previous results. For EV1, the variation range of SCs during the two years was rather small. The maximum variation of B for EV3, which underwent major repairs and component replacement, was up to 0.54 μT in the acceleration mode (difference between 2019SC1 and 2017SC1 for front seat position, refer to Table 3), whereas the maximum variation for EV1, which only underwent regular maintenance, was approximately 0.0.02 μT (difference between 2019SC1 and 2017SC1 for front seat position, refer to Table 3). EV2, which had its hubs changed, had a variation between those of EV1 and EV3. The results for the constant-driving mode revealed the same effect. In general, the frequency domain results were consistent. ”

CONCLUSION

To be able to support the thesis that the changes in B-field after repair and change of tires it would be good to support this with info on repair for each EV, for instance in Table 1.

--We made a supplementary table to summarize the SCs for each year by adding Table 3.

The spectral component is not significantly changed since no statistical analyses have been applied on this comparison.

--We did not conduct statistical for SC because it is hard to define a threshold for comparison. The shift in frequency may be also due to many factors involving in car driving besides of repair. However, the higher variation could be detected for the vehicles with major repairs and component’s replacement. We agree with you and we deleted the expression “significant (significantly)” when we described the difference for spectral components.

Round 3

Reviewer 3 Report

In the second revision, the authors have clarified most of my previous concerns. There are, however, still some issues that needs to be improved.

LANGUAGE: I still have some difficulties with the wording "result" B result or ELF MF result. For instance, line 131: "... and the measured B ELF MF results" is strange. If you mean ..."and the measured magnetic flux density" you could perhaps define an acronym to magnetic flux density (B-field or MFD) and use it through the text. There are many other similar wordings through out the text, please check carefully!

RESULTS: I can't see any reason for showing the raw data in Table 2.

The analyses of the spectral components are still vague. Figure 3 is still almost impossible to read, the dots are so big that they overlap each other and comparisons are impossible. Could you use thin lines instead?

The new Table 3 is ambitious, but since the frequency of  SC1-SC3 varies between years, no conclusions can be drawn. Would it be enough with a clear Figure 3 where the reader can visually inspect the spectrum of the EV for each measured year?  Please revisit you hypothesis and present the results accordingly. 

CONCLUSIONS

The difference in the spectral components have not been tested, but the magnetic flux density is, please clarify.

Author Response

In the second revision, the authors have clarified most of my previous concerns. There are, however, still some issues that needs to be improved.

--Thanks for your suggestion. We have corrected the manuscript accordingly.

LANGUAGE: I still have some difficulties with the wording "result" B result or ELF MF result. For instance, line 131: "... and the measured B ELF MF results" is strange. If you mean ..."and the measured magnetic flux density" you could perhaps define an acronym to magnetic flux density (B-field or MFD) and use it through the text. There are many other similar wordings through out the text, please check carefully!

--Thanks for your careful review. It was a typing error. B should be deleted and we meant to say “and the measured ELF MF results”. We used abbreviation (B) for magnetic flux density. We hope that it would make the manuscript much clearer and we update the manuscript thoroughly.

RESULTS: I can't see any reason for showing the raw data in Table 2.

-- We deleted the raw data in Table according to your suggestion.

The analyses of the spectral components are still vague. Figure 3 is still almost impossible to read, the dots are so big that they overlap each other and comparisons are impossible. Could you use thin lines instead?

-- We changed them to thin line dot. We hope that it looks better.

The new Table 3 is ambitious, but since the frequency of SC1-SC3 varies between years, no conclusions can be drawn. Would it be enough with a clear Figure 3 where the reader can visually inspect the spectrum of the EV for each measured year?  Please revisit you hypothesis and present the results accordingly. 

-- You have reason. We deleted the Table 3. Figure 3 and Figure 4 were re-drawn by thin-type dots. We hope that the visualization was improved.

CONCLUSIONS

The difference in the spectral components have not been tested, but the magnetic flux density is, please clarify.

-- Thanks for your valuable advice. We understood your concerns. Significant change was only used for describing the broadband B results. The paragraph was changed to “In this study, we monitored ELF MF in three shared vehicles over two years. The measurements were performed at the front and the rear seats under acceleration and constant-driving modes. We found that the broadband B value was significantly changed with replacement of the components and the tires while regular checks or maintenance did not influence the measured B values in the vehicle. The variation of the major spectral components of B was larger for the repaired cars, compared to the results from the cars with regular maintenance. The results highlight the necessity to regularly monitor the ELF MF in EVs, especially after major repairs or accidents, to protect car users from MF exposure.”

Round 4

Reviewer 3 Report

The authors have improved the manuscript in line with my previous raised comments. I would prefer thin lines instead of dots in Figure 3 to make it easier to interpret, but I leave this decision to the editor.